# Advancing Sparse Attention in Spiking Neural Networks via Spike-Timing-Based Prioritization

## Abstract

Current Spiking Neural Networks (SNNs) underutilize the temporal dynamics inherent in spike-based processing, relying primarily on rate coding while overlooking precise timing information that provides rich computational cues. We address this by proposing **SPARTA** (Spiking Priority Attention with Resource-Adaptive Temporal Allocation), which leverages heterogeneous neuron dynamics and spike-timing information to enable sparse attention mechanisms. SPARTA extracts temporal cues—including firing patterns, spike timing, and inter-spike intervals—to prioritize tokens for processing, achieving 65.4% sparsity through competitive gating. By selecting only the most salient tokens, SPARTA reduces attention complexity from $O(N^2)$ to $O(K^2)$, where $k \ll n$. Our approach achieves state-of-the-art accuracy on DVS-Gesture (98.78%) and competitive performance on CIFAR10-DVS (83.06%) and CIFAR-10 (95.3%), demonstrating that spike-timing utilization enables both computational efficiency and competitive accuracy.

## 1 Introduction

Deep learning with Artificial Neural Networks (ANNs) has revolutionized numerous aspects of modern society, achieving major breakthroughs in computer vision He et al. (2016). However, the substantial energy consumption of these increasingly complex models has emerged as a critical bottleneck for practical deployment Strubell et al. (2019).

Spiking Neural Networks (SNNs) offer a fundamentally different paradigm by processing discrete, asynchronous spikes that mirror biological neural computation. This enables natural event-driven processing and intrinsic temporal coding, making SNNs particularly suited for neuromorphic hardware such as Intel's Loihi Davies et al. (2018). Yet, the discrete spike events and temporal dynamics introduce inherent challenges for training, resulting in a persistent accuracy gap compared to continuous-valued ANNs. While recent approaches have successfully improved SNN performance through ANN-inspired techniques like ANN-to-SNN conversion Deng & Gu (2021), surrogate gradient methods Neftci et al. (2019), and sophisticated normalization Zheng et al. (2021), these methods have underutilized the rich temporal dynamics inherent in spike-based processing. Most existing approaches focus primarily on achieving higher accuracy while insufficiently exploiting the precise temporal information that distinguishes SNNs from conventional ANNs, representing a key opportunity for further advancement. Consequently, this has limited their computational efficiency on neuromorphic hardware Orchard et al. (2021); Bellec et al. (2018).

SNNs inherently excel in spatio-temporal coding, leveraging precise spike timing to efficiently encode complex temporal patterns, a capability ANNs do not naturally replicate Eshraghian et al. (2023). However, current research predominantly focuses on matching ANN performance metrics while insufficiently addressing the unique opportunities that spike timing provides for both computational efficiency and attention mechanisms. This raises a critical question: *How can we more effectively integrate the temporal coding capabilities inherent to spike dynamics into attention mechanisms to enhance both efficiency and performance in SNNs?*

Inspired by neuroscientific evidence from insights into temporal dynamics underlying selective attention Singer (1999), we propose SPARTA (**S**piking **P**riority **A**ttention with **R**esource-Adaptive **T**emporal **A**llocation). SPARTA incorporates heterogeneous neurons that mimic the rich diversity of

response properties found in cortical neuron populations, enabling the network to capture complex temporal features across multiple timescales. It then leverages a Spatio-Temporal Encoding Network (STEN) to construct a multi-scale feature representation that explicitly preserves this critical spike timing information. Finally, these features guide a Priority-Aware Sparse Temporal Attention, which dynamically allocates computational resources only to the most salient tokens, avoiding the quadratic complexity of standard attention while maintaining processing efficiency.

SPARTA's token selection mechanism is based on three biologically-inspired observations: (1) important stimuli tend to fire earlier Foffani et al. (2009), (2) important stimuli tend to fire with shorter intervals between spikes Oswald et al. (2007), and (3) important stimuli tend to fire more frequently Gerstner et al. (1997). These observations guide our multi-scale feature extraction approach. We interpret stimuli as tokens, enabling biologically-inspired selection and localized competitive gating for sparse attention, while preserving event-driven sparsity critical to neuromorphic hardware.

- **HI-LIF neuron** (Heterogeneous Initialized Leaky Integrate-and-Fire) that introduces learnable, channel-wise temporal diversity to expand the network's processing bandwidth.
- A novel **sparse attention mechanism** guided by biologically-inspired temporal cues (firing rate, spike timing) for efficient, saliency-based computation.
- The **SPARTA framework**, which achieves competitive performance demonstrating that integrating biologically-inspired temporal cues can enhance both efficiency and performance in SNNs.

## 2 BACKGROUND AND MOTIVATION

### 2.1 LIF NEURON MODELS AND TEMPORAL CODING

The computational core of Spiking Neural Networks (SNNs) is the Leaky Integrate-and-Fire (LIF) neuron, whose dynamics are governed by a membrane time constant ($\tau$) and a firing threshold ($v_{th}$), as described in Equations 1 and 2.

$$u^{(t+1)} = u^{(t)} \left(1 - \frac{1}{\tau}\right) + x^{(t)} \tag{1}$$

$$s^{(t)} = \Theta \left(u^{(t)} - v_{th}\right) \tag{2}$$

However, a critical limitation arises in how these models are conventionally applied: SNNs typically employ **uniform parameters** shared across all spatial channels. This simplification, while computationally convenient, starkly contrasts with biological reality, where cortical neurons exhibit remarkable **heterogeneity** in their temporal properties Mason et al. (2022); Eyal et al. (2023). This imposed homogeneity creates a significant bottleneck, limiting the network's temporal coding capacity and its ability to process complex information across multiple timescales Perez-Nieves et al. (2021). Our work directly confronts this limitation, proposing a neuron model inspired by this biological diversity to unlock a richer temporal processing bandwidth.

### 2.2 HUMAN COGNITION AND TEMPORAL ATTENTION

Human visual attention leverages temporal dynamics for rapid pattern recognition. The flashed-face distortion effect demonstrates temporal sensitivity: faces presented in rapid succession appear perceptually distorted, consistent with competitive normalization within brief presentation windows Tangen et al. (2011). Structured visual search tasks illustrate temporal attention mechanisms where systematic scanning operates under specific temporal constraints Wolfe (1994); Chun & Potter (1995). Cognitive integration depends on maintaining partial cues within critical time windows, beyond which integration success declines Chun & Potter (1995); Di Lollo (1977). While these cognitive phenomena provide a high-level, conceptual foundation for our approach, the core computational mechanisms of SPARTA are grounded in established models from computational neuroscience.

**The First Word You Find Describes
Your Psychological State !**

Figure 1: Crossword puzzle analogy illustrating temporal integration and decay of visual cues underlying SPARTA's selective temporal attention.

Figure 1 illustrates a step-by-step temporal attention process analogous to our problem:

1. **Global scan:** Continuous monitoring detects "FAT" but dismisses it, as "FAT" is not a valid psychological word.

2. **Rapid detection:** Focused attention shifts to "IGUE" within the relevant time interval.

3. **Temporal integration:** The brain associates "FAT" and "IGUE" to form "FATIGUE" (a valid psychological term).

4. **Interval dependency:** Longer delays cause the memory of "FAT" to decay, reducing the likelihood of integration. The user will be unable to connect "IGUE" with "FAT".

Such observations suggest how temporal windows may affect the selection and integration of sensory cues, providing a conceptual foundation for SPARTA's spike-timing based attention mechanism. This temporal integration process parallels spike-based neural computation, where information binding occurs through precise timing relationships rather than simple accumulation. Unlike conventional frame-based approaches that process each temporal snapshot independently, spiking networks naturally maintain temporal context through membrane dynamics, enabling information integration across biologically plausible time windows.

Building on this principle, SPARTA leverages spike timing to preserve the temporal dynamics essential for efficient attention allocation, drawing inspiration from both cognitive processes and neuromorphic computation principles.

### 2.3 EVENT-BASED VISION

Event-based cameras (e.g., Dynamic Vision Sensors; DVS) emit polarity events only when local log-intensity changes occur, producing temporally precise and highly sparse streams that interface cleanly with spike-based SNN computation. Lichtsteiner et al. (2008); Gallego et al. (2020). These sensors capture key aspects of change-driven encoding found in biological vision systems Delbruck & Lichtsteiner (2014); Gollisch & Meister (2010). Learning directly from native event streams is practical at scale; accumulating events into dense frames discards fine-grained timing info and can substantially increase memory and computational demands Amir et al. (2017); Gallego et al. (2020). Neuromorphic processors (e.g., Loihi 2) achieve substantial energy savings by scheduling work only on active addresses—idle pixels and silent neuron populations incur negligible cost—making event-driven sparsity a primary efficiency lever Davies et al. (2018); Orchard et al. (2021). Accordingly, SPARTA retains temporally resolved spike tokens and applies salience-guided sparsification prior to global attention, focusing compute on active regions.

## 3 RELATED WORKS

**Performance-Driven SNNs** focus on accuracy through ANN-inspired techniques, primarily emphasizing rate-based coding. ANN-to-SNN conversion methods achieve high performance through weight scaling and threshold adjustment Bu et al. (2022); Han et al. (2020); Kim et al. (2022). Hybrid training schemes apply gradient-based optimization to spike networks Li et al. (2021). Large-scale adaptations port state-of-the-art models to spiking paradigms Yao et al. (2023); Zheng et al. (2023); Wang et al. (2023), demonstrating compatibility with cutting-edge AI while treating spikes primarily as discrete rate codes rather than exploiting their temporal coding capabilities.

**Biologically-Inspired Architectures** emphasize neuromorphic principles and biological similarity in design. Local learning rules employ STDP and Hebbian mechanisms for biological fidelity Song et al. (2000); Diehl & Cook (2015), offering hardware compatibility while optimizing for specific learning paradigms that prioritize biological authenticity. Neuromorphic hardware designs achieve energy efficiency and real-time processing through platform-specific optimizations for systems like Loihi Davies et al. (2018) and SpiNNaker Furber et al. (2014), demonstrating effective integration of neuromorphic principles with practical hardware constraints. Brain-circuit architectures directly emulate neural circuits through cortical column simulations Hawkins & Ahmad (2016), achieving high biological similarity while specializing in applications where biological fidelity is the primary design criterion.

**Temporal Dynamics & Sparse Processing** focuses on preserving and leveraging spike timing information to optimize computational efficiency and accuracy. Methods like optimized spiking neurons achieve high accuracy through precise timing codes Stockl & Maass (2021), while sparse processing frameworks reduce energy consumption without sacrificing network performance Yin et al. (2023). Multi-scale encoding approaches expand receptive fields across resolution levels Dampfhoffer et al. (2021), and progressive learning methods enable deep networks to process complex patterns through sparse representations Wu et al. (2021). Recent empirical analysis demonstrates temporal information dynamics in SNNs, showing natural concentration in earlier timesteps during training Kim et al. (2023). Spatio-temporal attention mechanisms effectively integrate temporal dependencies without additional computational overhead Lee et al. (2025). While these approaches demonstrate significant progress in leveraging temporal information, there remains opportunity to further integrate biological principles with sparse attention mechanisms to achieve balance between temporal coding and computational efficiency.

## 4 METHODOLOGY

### 4.1 HI-LIF: HETEROGENEOUS TEMPORAL DYNAMICS.

Real cortical neurons exhibit variability in membrane time constants ($\tau$) and firing thresholds ($v_{th}$) Mason et al. (2022); Eyal et al. (2023). We implement channel-wise initialization for LIF neurons, sampling $\tau$ and $v_{th}$ per channel from learnable normal distributions. This *Heterogeneous Initialized LIF* (HI-LIF) introduces temporal diversity across channels.

Channel-wise diversity in $\tau_{\text{inv}}^{(c)}$ and $v_{th}^{(c)}$ yields a spectrum of rapid and sluggish responders: low-$\tau$/low-$v_{th}$ paths fire early to encode high-frequency events, whereas high-$\tau$ paths integrate slow dynamics. This dual heterogeneity broadens the network's temporal receptive field while preserving its overall computational efficiency, enabling the model to capture both fleeting transients and long-range context within the event-driven layer.

### 4.2 MULTI-SCALE SPIKE ENCODING & FEATURE EXTRACTION

As mentioned in introduction, SPARTA's token selection pipeline ranks patch saliency using three biologically driven cues—*high firing rate*, *early first-spike*, and *short inter-spike interval*.

#### 4.2.1 SPATIO-TEMPORAL ENCODING NETWORK(STEN)

STEN implements cascading downsampling that preserves temporal dynamics through HI-LIF neurons. It processes spike features through three parallel branches: (i) 1×1 convolution for fine-grained

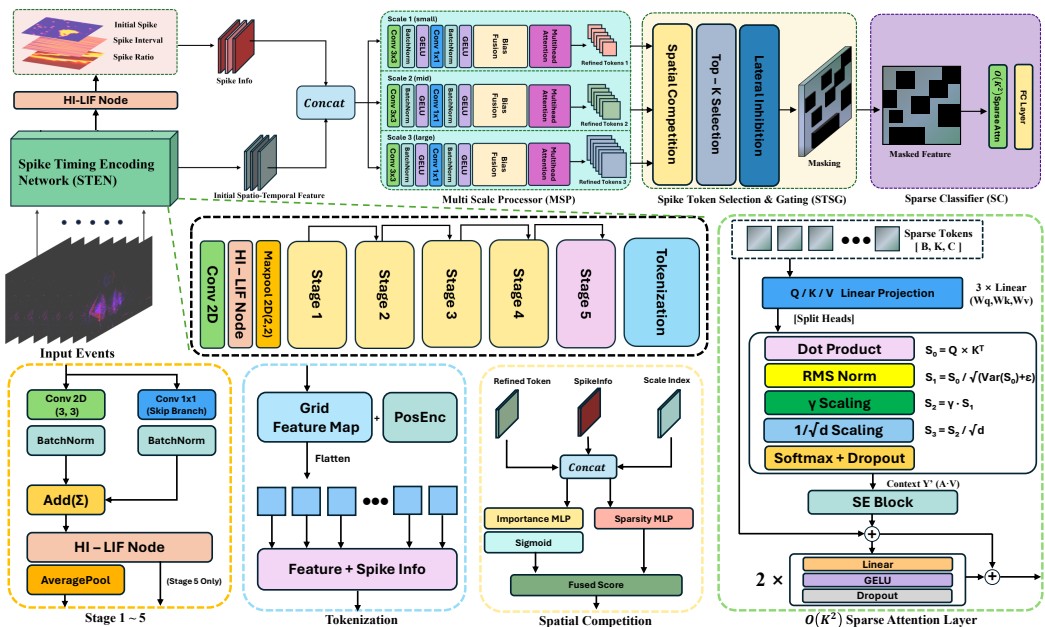

Figure 2: SPARTA architecture overview. Input events are processed through STEN to generate SpikeInfo with rich temporal cues. Following multi-scale processing by MSP, the STSG module modulates token importance, after which the SC selects the top-K salient tokens for efficient sparse attention classification. Solid arrows show the main data flow.

details, (ii) 3×3 convolution with HI-LIF for additional dynamics, and (iii) adaptive pooling for global context. The resulting multi-scale features are concatenated and refined by a timing-aware attention mechanism, which assigns higher weights to rich temporal activity. In parallel, STEN derives three complementary timing metrics—first-spike timing $T_{\text{first}}$ for rapid event detection, inter-spike intervals $T_{\text{interval}}$ for continuity, and burst firing patterns $T_{\text{burst}}$ for salience estimation, forming a comprehensive temporal representation for downstream processing.

### 4.2.2 MULTI-SCALE PROCESSING

The second stage applies bias-based attention mechanisms that weight different temporal characteristics. This approach is grounded in established temporal coding models from computational neuroscience. Specifically, we model the decaying importance of spike latency using an exponential function, a common and effective method in temporal plasticity models. The weights are defined as:

$$w_{\text{timing}}^{(s)} = \exp(-\alpha \cdot T_{\text{first}}^{(s)}) \tag{3}$$

$$w_{\text{interval}}^{(s)} = \exp(-\beta \cdot T_{\text{interval}}^{(s)}) \tag{4}$$

$$w_{\text{combined}}^{(s)} = w_{\text{timing}}^{(s)} \odot w_{\text{interval}}^{(s)} \odot \sigma(\gamma \cdot F_{\text{rate}}^{(s)}) \tag{5}$$

Here, the exponential formulation for $w_{\text{timing}}^{(s)}$ directly models the principle that earlier spikes carry greater informational value, a key aspect of first-spike latency codes. The factors $\alpha, \beta, \gamma$ are learnable scaling parameters that allow the network to adaptively balance these complementary temporal cues. The resulting temporal bias is incorporated into multi-head attention through attention masking to emphasize temporally salient regions.

### 4.2.3 PATCH GROUPING

Aggregates multi-scale spike features and adjusts the token count to a fixed size without zero-padding by selecting tokens based on their importance (e.g., firing rate). This preserves meaningful spike information while ensuring compatibility with downstream modules.

## 4.3 SPARSE TOKEN PROCESSING & ATTENTION

The sparse processing stage implements biologically-inspired competition and selective attention through integrated mechanisms that reduce computation while preserving salient temporal patterns. By dynamically selecting the top-$k$ tokens where $k \ll n$, it achieves an efficient attention complexity of $\mathcal{O}(k^2)$, significantly lowering the computational cost compared to the full $\mathcal{O}(n^2)$ attention.

### 4.3.1 SPIKE TOKEN SELECTION & GATING (STSG).

STSG implements lateral inhibition mechanisms that integrate three attention mechanisms: MSP features, spatial competition through center-surround inhibition kernels, and temporal priority information. Unlike fixed sparsity ratios, STSG employs a learned predictor that adapts to temporal characteristics:

$$\mathbf{f}_{\text{input}} = \left[ \begin{array}{c} \text{mean}(\mathbf{F}_{\text{rate}}) \\ \text{std}(\mathbf{T}_{\text{timing}}) \\ \text{mean}(\mathbf{T}_{\text{interval}}) \end{array} \right] \tag{6}$$

The three scoring mechanisms are fused through a learned attention network:

$$\mathbf{s}_{\text{combined}} = \text{MLP}_{\text{fusion}}([\mathbf{s}_{\text{spatial}}, \mathbf{s}_{\text{MSP}}, \mathbf{s}_{\text{temporal}}]) \tag{7}$$

The dynamic K value is computed based on predicted sparsity ratio with a minimum threshold for stable processing, and top-K tokens are selected based on fused attention scores. The lateral inhibition mechanism applies differential processing where selected tokens receive enhancement while non-selected tokens undergo suppression:

$$\mathbf{f}_{\text{processed}} = \mathbf{f} \odot \mathbf{M}_{\text{topK}} \cdot \alpha + \mathbf{f} \odot (\mathbf{I} - \mathbf{M}_{\text{topK}}) \cdot \beta \tag{8}$$

where $\mathbf{M}_{\text{topK}}$ is the binary selection mask, $\mathbf{I}$ is the identity matrix, and $\alpha, \beta$ are learned enhancement and suppression factors respectively.

### 4.3.2 SPARSE ATTENTION CLASSIFIER (SC).

The final module receives the temporally-modulated tokens from the STSG and implements genuine, hard sparsity by selecting content-adaptive top-$k$ tokens based on temporal urgency. It processes only this reduced set for classification, reducing computational complexity from $O(N^2)$ to $O(K^2)$ where $K \ll N$. Token selection uses a dynamic priority score that synthesizes three biologically-motivated cues from the STSG output: early-firing tokens receive higher priority (rapid stimulus detection), tokens with shorter inter-spike intervals gain precedence (sustained attention), and higher firing rates indicate stimulus salience. A temporal integration network mimics cortical attention circuits to select tokens with the most significant temporal patterns.

### 4.3.3 O($k^2$) SPARSE ATTENTION LAYER.

The selected $k$ tokens are processed through specialized attention layers that perform Priority-Aware Sparse Temporal Attention, adapting focus based on temporal characteristics. Early-firing tokens with short inter-spike intervals receive sharper attention allocation, while tokens with delayed or irregular firing patterns are processed with broader attention distributions. This temporal adaptation concentrates computational resources on time-critical information, mirroring biological selective attention mechanisms. Multi-layered processing enables hierarchical refinement of temporal priorities, preserving the most salient temporal dynamics for classification.

## 5 EXPERIMENTS

To validate the effectiveness of our approach, we conduct experiments on neuromorphic datasets (DVS Gesture Amir et al. (2017), CIFAR10-DVS Li et al. (2017)) and conventional RGB datasets (CIFAR-10 Krizhevsky & Hinton (2009), CIFAR-100 Krizhevsky & Hinton (2009)). Our evaluation focuses on analyzing SPARTA's overall performance and temporal variance of model.

**Experimental Setup.** Experiments were performed using PyTorch with AdamW, CrossEntropy Loss (lr 1e-4, cosine schedule); neuromorphic tasks ran 300 epochs, RGB tasks 500 epochs. Results are the mean of three seeds.

### 5.1 STUDY ON TEMPORAL RESOLUTION.

We first analyze SPARTA's performance across different time step configurations to understand its temporal processing characteristics and establish optimal operating conditions.

Table 1: Temporal performance at different timesteps. The variance column reports the spatial variance of first-spike timings and inter-spike intervals, respectively, computed across all output tokens and averaged over the entire test set.

| Timesteps | Accuracy (%) | | Variance |
|---|---|---|---|
| | DVS-Gesture | CIFAR10-DVS | Timing / Interval |
| 4 | 88.89 (-10.01%) | 78.2 (-6.07%) | 0.287 / 1.81 |
| 8 | 92.74 (-6.04%) | 78.7 (-5.45%) | 0.288 / 1.93 |
| 12 | 94.73 (-4.05%) | 81.90 (-1.45%) | 0.280 / 1.92 |
| 16 | 98.46 (-0.32%) | **83.06** | 0.288 / 1.71 |
| 20 | **98.78** | 82.87 (-0.24%) | 0.292 / 1.82 |
| 32 | 94.30 (-4.54%) | 79.45 (-4.51%) | 0.287 / 1.91 |

Results show optimal performance at T=20 for DVS-Gesture (98.78%) and T=16 for CIFAR10-DVS (83.06%). Both datasets exhibit performance degradation at T=32 (-4.54% and -4.51%), indicating that excessive temporal windows introduce noise and reduce accuracy.

### 5.2 HI-LIF HETEROGENEITY ANALYSIS.

We analyze the impact of channel-wise heterogeneity on temporal encoding efficacy by varying the standard deviations of the membrane time constant ($\tau$) and firing threshold ($v_{th}$).

Table 2: Impact of individual HI-LIF parameter diversity on accuracy (DVS-Gesture) ($\tau_\mu = 2.0$, $v_{th,\mu} = 1.0$, T=16).

| Configuration | HI-LIF Parameters | | Accuracy |
|---|---|---|---|
| | $\tau_\sigma$ | $v_{th,\sigma}$ | (%) |
| Homogeneous | 0.0 | 0.0 | 96.53 |
| Low Tau Diversity | 0.2 | 0.0 | 97.25 |
| High Tau Diversity | 0.5 | 0.0 | 94.36 |
| Low Threshold Diversity | 0.0 | 0.1 | 96.27 |
| High Threshold Diversity | 0.0 | 0.3 | 92.43 |
| Combined Diversity | 0.5 | 0.3 | 89.83 |
| **Adjusted (Combined)** | 0.3 | 0.2 | **98.46** |

Results show that adjusted combined heterogeneity achieves the highest accuracy, demonstrating the benefit of balanced diversity, while excessive diversity leads to performance degradation due to instability.

### 5.3 ANALYSIS OF SPARSITY POLICIES.

We benchmark our dynamic sparsity policy against fixed-sparsity baselines on the DVS-Gesture dataset ($N = 256$), evaluating the trade-off between accuracy and computational cost.

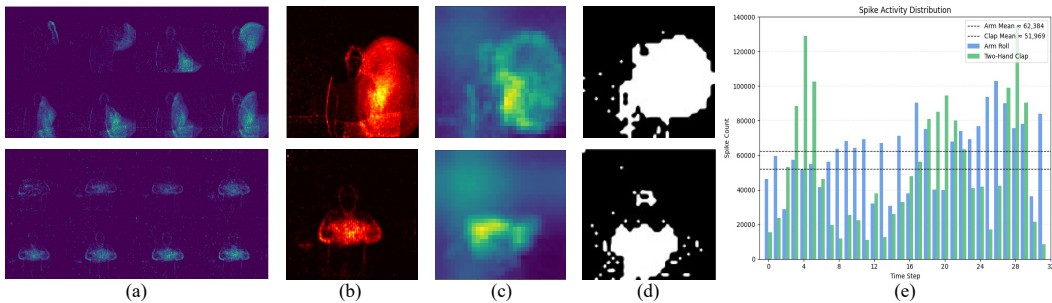

| (a) | (b) | (c) | (d) | (e) |

Figure 3: Visualization of gesture samples.**(a)** Input frame from DVS-Gesture dataset (top: arm-roll, bottom: clap gestures); **(b)** Firing rate map (red to yellow: low to high rates); **(c)** Attention weights (purple to yellow: low to high); **(d)** Top-K selection mask (white: selected tokens, black: filtered tokens); **(e)** Spike count variance across temporal dimension. The *arm-roll* gesture shows uniform spike counts over time, while *clap* exhibits concentrated spike bursts.

Table 3: DVS-Gesture accuracy and computational cost for dynamic vs. fixed sparsity policies (T=20). For fixed policies, K denotes the number of tokens resulting from the specified sparsity, calculated as $N \times (1 - \text{Sparsity}/100)$.

| Sparsity Policy | Sparsity (%) | Acc. (%) | FLOPs (G) |
|---|---|---|---|
| Dynamic (Ours) | 65.4 (Adaptive) | **98.78** | 1.23 |
| Fixed (K=192) | 25.0 | 98.30 | 1.24 |
| Fixed (K=128) | 50.0 | 92.50 | 1.21 |
| Fixed (K=64) | 75.0 | 78.03 | **1.19** |

Our dynamic policy achieves 65.4% sparsity while maintaining 98.78% accuracy, outperforming fixed baselines (Table 3). Unlike fixed approaches that trade accuracy for computational cost, our learnable predictor adapts token selection ($K$) to input complexity.

### 5.4 MSP TEMPORAL WEIGHTING ABLATION.

We systematically ablate the temporal weighting parameters in MSP to understand the contribution of each biological cue.

Table 4: Ablation study on MSP's temporal weighting cues (Accuracy %). $\alpha$, $\beta$, and $\gamma$ correspond to the weights for first-spike timing, inter-spike interval, and firing rate, respectively.

| Configuration | T | DVS-Gesture | CIFAR10-DVS |
|---|---|---|---|
| Full MSP ($\alpha,\beta,\gamma$) | 16 | **98.46** | **83.06** |
| w/o $\alpha$ (timing) | 16 | 96.94 (-1.52) | 79.80 (-3.26) |
| w/o $\beta$ (interval) | 16 | 95.56 (-2.90) | 80.35 (-2.71) |
| w/o $\gamma$ (firing rate) | 16 | 92.26 (-6.2) | 77.23 (-5.83) |
| w/o $\alpha,\beta$ | 16 | 94.61 (-3.85) | 78.28 (-4.78) |
| w/o $\alpha,\gamma$ | 16 | 89.16 (-9.30) | 75.26(-7.80) |
| w/o $\beta,\gamma$ | 16 | 91.38 (-7.08) | 74.50 (-8.56) |
| w/o $\alpha,\beta,\gamma$ | 16 | 85.0 (-13.46) | 72.80 (-10.26) |

Table 4 reveals a clear hierarchy of importance among the temporal cues. The firing rate weight ($\gamma$) is consistently the most critical single factor, as its removal causes the largest individual performance drop on both DVS-Gesture (-6.2%) and CIFAR10-DVS (-5.83%). While timing ($\alpha$) and interval ($\beta$) cues also contribute significantly, the most substantial degradation occurs when all three are removed entirely. This confirms that these weights work in a complementary manner to effectively guide the attention mechanism.

## 5.5 TEMPORAL VARIANCE ANALYSIS.

We analyze SPARTA's spike timing variance against other SNN architectures by measuring the spatial variance of first-spike timings and inter-spike intervals across output tokens, averaged over the DVS-Gesture test set Perez-Nieves et al. (2021); Bellec et al. (2018); Stockl & Maass (2021); Mason et al. (2022).

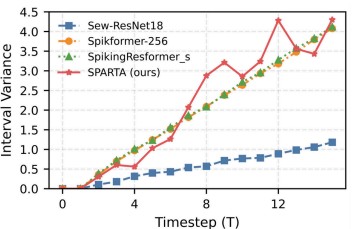 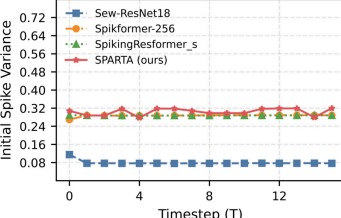

Figure 4: Temporal variance comparison across different SNN architectures on the DVS-Gesture. The y-axis represents the spatial variance of spike timings and intervals, averaged over the test set.

As shown in Figure 4, SPARTA maintains higher temporal diversity than the compared models. This confirms that the heterogeneous parameters of our HI-LIF neurons enable the network to learn richer and more diverse temporal representations, which is a key factor in its strong performance.

## 5.6 COMPARISON WITH STATE-OF-THE-ART

We conduct comprehensive experiments to compare our proposed SPARTA method with recent state-of-the-art SNN models across four benchmark datasets. Table 5 presents the performance comparison on both neuromorphic datasets (DVS-Gesture, CIFAR10-DVS) and static RGB datasets(CIFAR-10, CIFAR-100).

Table 5: State-of-the-art comparison on neuromorphic and RGB datasets. T denotes the timesteps; Acc (%) indicates classification accuracy; Params refers to the number of model parameters. A dash (–) indicates values not reported in the original paper.

| Method | Params (M) | DVS-Gesture | | CIFAR10-DVS | | CIFAR-10 | | CIFAR-100 | |
|---|---|---|---|---|---|---|---|---|---|
| | | T | Acc | T | Acc | T | Acc | T | Acc |
| SEW-ResNet | 60.2 | 16 | 89.06 | 16 | 67.20 | - | - | 4 | 75.93 |
| GLIF+ResNet | 11.2 | - | - | 16 | 78.10 | 4 | 94.67 | 4 | 77.37 |
| Spikformer | 9.32 | 16 | 95.49 | 16 | 80.60 | 4 | 95.19 | 4 | 77.86 |
| SpikingResFormer | 17.25 | 16 | 91.67 | 10 | 84.80 | 4 | 97.40 | 4 | 85.98 |
| QKFormer-S/L | 1.5/6.74 | 16 | 98.60 | 16 | 84.00 | 4 | 96.18 | 4 | 81.15 |
| SGLFormer | 8.9 | 16 | 97.20 | 10 | 82.90 | 4 | 96.76 | 4 | 82.26 |
| Event-Vivid | 48.2 | 20 | 98.80 | 20 | 92.50 | - | - | - | - |
| SMA-AZO-VGG | - | 16 | 98.60 | 10 | 84.00 | - | - | - | - |
| SPARTA (Ours) | 13.8 | 20 | **98.78** | 16 | 83.06 | 4 | 95.3 | 4 | 78.1 |

## 6 CONCLUSION

We present SPARTA, which integrates spike-timing information into sparse attention mechanisms for SNNs. The framework uses channel-wise heterogeneous initialization and biologically-motivated temporal cues to achieve $O(K^2)$ attention complexity with 65.4% sparsity.

Our results on DVS-Gesture (98.78%) and CIFAR10-DVS (83.06%) validate that biological principles can coexist with high accuracy in large-scale SNNs. However, the sparse attention mechanism may lead to information loss when critical information is distributed across many tokens (e.g., fine-grained textures) or when important temporal patterns occur in tokens filtered out during top-K selection. Future work will focus on extending SPARTA to multi-modal data and deploying the framework on neuromorphic hardware to verify real-world efficiency.

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
