# OpenReview forum: "Advancing Sparse Attention in Spiking Neural Networks via Spike-Timing-Based Prioritization"
_ICLR.cc/2026/Conference — ICLR 2026 Conference Withdrawn Submission_

### Official Review · Reviewer_Gv7r · 2025-10-26

**Soundness:** 1
**Presentation:** 1
**Contribution:** 1
**Rating:** 2
**Confidence:** 5

**Summary:**

This paper proposed the Spiking Priority Attention with Resource Adaptive Temporal Allocation (SPARTA) to enable sparse attention mechanisms.

**Strengths:**

Clear motivation, but I think the paper would benefit from significant revision.

**Weaknesses:**

The authors adopted a notably complex method to assess the temporal characteristics of spike trains, followed by another intricate approach for token selection. Based on the description in the Methods section and the provided code, it seems that the proposed “token selection” procedure is computationally more expensive than directly performing the spike calculation itself. However, the authors only take the sparsity/flops of the model as the effectiveness metric of their proposed method.

While authors claim that SPARTA achieves better performance due to enhanced temporal diversity, results show that accuracy decreases with larger models and longer simulation steps. Furthermore, although the paper focuses on sparse attention, all evaluations are conducted on relatively small datasets.  This raises concerns about the generalizability of the proposed approach. It is also unclear why the authors characterize their model as “state-of-the-art” despite its inferior results on CIFAR10-DVS, CIFAR10, and CIFAR100 benchmarks.

While “energy efficiency” and “sparsity” are emphasized as the core advantages of this work, Appendix A.3 shows that the proposed model exhibits substantially higher energy consumption compared with recent approaches evaluated on ImageNet [1]. In addition, many of the figures are difficult to interpret and do not meet the presentation quality expected for ICLR submissions.

Due to the above concerns, I do not recommend acceptance at this stage.

[1] Spiking Transformers Need High-Frequency Information. Neurips 2025.

**Questions:**

See weakness.

---

### Official Review · Reviewer_Rywp · 2025-10-26

**Soundness:** 2
**Presentation:** 2
**Contribution:** 2
**Rating:** 4
**Confidence:** 4

**Summary:**

This paper introduces SPARTA, a biologically-inspired sparse attention framework for SNNs. It leverages precise spike-timing cues (first-spike time, inter-spike interval, and firing rate) to prioritize and selectively process tokens, aiming to improve both accuracy and efficiency. The method also incorporates heterogeneous neurons (HI-LIF) to enhance temporal feature representation.

**Strengths:**

1. The core idea of using precise spike-timing information, beyond just firing rates, to guide the attention mechanism in SNNs is highly original and significant. It aligns well with the principles of neuromorphic computing and opens a new avenue for designing more brain-like attention.

2. The method achieves state-of-the-art accuracy on the DVS-Gesture dataset, demonstrating that the proposed timing-based prioritization can be very effective for tasks with rich and distinct temporal dynamics.

**Weaknesses:**

1. The proposed SPARTA framework is exceedingly complex. It introduces multiple specialized modules (STEN, MSP, STSG, SC) and a cascade of processing steps simply to select which tokens to attend to. The energy breakdown in the appendix reveals that the vast majority of energy (83.7%) is consumed by the front-end STEN module, suggesting that the feature extraction process itself is a major bottleneck. This complexity makes the model difficult to reproduce, analyze, and scale, and it goes against the goal of designing simple and efficient neural architectures. The lack of crucial metrics like inference latency further obscures a true assessment of its practical efficiency.

2. Despite its focus on sparsity, the model's energy consumption is not low in absolute terms. The reported energy on DVS-Gesture is higher than recent SOTA SNN Transformers (e.g., Spiking Wavelet Transformer), which achieve better performance on more difficult tasks with only ~3.5 mJ. The paper's claim of "computational efficiency" is therefore misleading. While it improves upon some older models, it is not competitive on the current state-of-the-art efficiency frontier.

3. The paper's experimental results are weak. Most importantly, it was not tested on ImageNet. On top of that, its performance on the smaller CIFAR datasets is not very good. It gets 95.3% on CIFAR-10, which is lower than other recent SNNs that also use less energy.

**Questions:**

1. Can you provide the performance results for SPARTA on the ImageNet dataset?

2. Could you provide a direct comparison and explain how SPARTA remains competitive in terms of the accuracy-efficiency trade-off against such state-of-the-art efficient spiking transformer architectures on the ImageNet dataset?

3. To fully evaluate the practical efficiency of your method, could you report the inference latency and compare it to other baseline models?

---

### Official Review · Reviewer_kHbY · 2025-10-28

**Soundness:** 2
**Presentation:** 2
**Contribution:** 2
**Rating:** 2
**Confidence:** 5

**Summary:**

This paper addresses the underutilization of precise temporal dynamics in Spiking Neural Networks (SNNs), which often rely primarily on rate coding. It proposes SPARTA (Spiking Priority Attention with Resource-Adaptive Temporal Allocation), a novel framework that leverages spike-timing information to enable an efficient sparse attention mechanism

**Strengths:**

The paper clearly identifies a key limitation in the SNN field: an over-reliance on rate-based coding that overlooks rich temporal information. It proposes a novel and well-justified method that uses multiple bio-inspired temporal cues (e.g., first-spike timing, inter-spike intervals, and firing frequency) to guide a sparse attention mechanism.

**Weaknesses:**

1. The paper's validation is limited to smaller-scale datasets (DVS-Gesture, CIFAR-10/100). It lacks experiments on large-scale benchmarks such as ImageNet, which is a standard requirement for validating the scalability and generalization capabilities of a novel architecture in the broader deep learning field.
2. Insufficient Comparison to SOTA SNN Encoding Methods: The related work section and SOTA comparison (Table 5)  are incomplete. The paper fails to discuss or benchmark against other relevant and advanced SNN strategies, such as state-of-the-art "direct coding" SNNs [1] or "GAC-SNN" [2]  encoding methods. This omission makes it difficult to assess the true novelty and advantage of SPARTA compared to other advanced techniques addressing SNN efficiency and encoding.

[1] Direct training for spiking neural networks: Faster, larger, better. AAAI 2021
[2] Gated attention coding for training high-performance and efficient spiking neural networks. AAAI 2024

**Questions:**

see weakness

---

### Official Review · Reviewer_Tvtr · 2025-10-31

**Soundness:** 2
**Presentation:** 1
**Contribution:** 2
**Rating:** 2
**Confidence:** 5

**Summary:**

This paper introduces SPARTA (Spiking Priority Attention with Resource-Adaptive Temporal Allocation), a framework that leverages spike-timing information to enable sparse attention mechanisms in Spiking Neural Networks (SNNs). The key contributions include HI-LIF ( Heterogeneous Initialized Leaky Integrate-and-Fire) neurons, Spatio-Temporal Encoding Network (STEN) that extracts three temporal cues,  Sparse attention (from $O(N^2)$ to $O(K^2)$ by dynamically selecting top-K salient tokens based on biologically-inspired temporal priorities.

**Strengths:**

1. Biological motivation
- The integration of three complementary temporal cues (early firing, short inter-spike intervals, high firing rate) is well-grounded in neuroscience literature and provides a principled approach to token prioritization.

2. Comprehensive experimental analysis
- Ablation studies on each component (HI-LIF heterogeneity, temporal weighting parameters)
- Analysis across different temporal resolutions
- Comparison of granularity levels (layer-wise, channel-wise, neuron-wise)

3. Temporal variance analysis
- The paper verifies that SPARTA maintains higher temporal diversity compared to other SNN architectures, validating the effectiveness of HI-LIF neurons.

**Weaknesses:**

1. Hard to read
- I couldn't follow the methodology section in conjunction with Figure 2. The figure appears disorganized, with substantial misalignment between the textual descriptions and visual representation. For example, in Section 4.2.1 (STEN), the authors mention 1×1 convolution and adaptive pooling, but these components are not identifiable in the figure. Additionally, it is unclear whether Section 4.2.3 (Patch Grouping) corresponds to STSG + SC or represents a separate component.


2. Figure 1
- Figure 1 does not appear to convey any core ideas or contributions essential to this paper. I cannot identify any empirical or theoretical results that demonstrate a meaningful correlation with the crossword puzzle analogy presented in Figure 1.


3. Scalability & Limited dataset
- This paper only experiments on small-scale datasets (CIFAR-10/100, DVS-Gesture) with a relatively small architecture (13.8M parameters). Current state-of-the-art Spiking Transformer works demonstrate scalability on large-scale datasets, with ImageNet serving as the minimum benchmark requirement.


4. Energy Consumption
- In the Appendix, the authors calculate energy consumption using $E_{AC}$ for all layers except the first convolutional layer. However, in Figure 2's Sparse Attention Layer, the inputs to linear layers and other modules do not appear to be binarized values. In this case, energy consumption should be calculated using $E_{MAC}$.

**Questions:**

1. HI-LIF

- Is there any computational or memory overhead introduced by using channel-wise $\tau$ and $v_{th}$​? Additionally, regarding Table 2, I am somewhat confused about the channel-wise $\tau$ and $v_{th}$​ configuration. Does the "Adjusted (Combined)" setting use constant values of 0.3 and 0.2 for $\tau$ and $v_{th}$, respectively?

2. Sparsity

- The authors claim that SPARTA achieves high sparsity at 65.4%. However, to my knowledge, many existing spiking architectures, including spiking transformers, typically achieve sparsity levels in the range of 60-80%. Could you clarify what makes your sparsity achievement particularly significant or different from these existing methods?


3. Computational overhead

- Your architecture incorporates many additional modules compared to other spiking transformers. Could you provide quantitative comparisons of training time and inference overhead relative to baseline architectures?

---

### Note · Authors · 2025-11-12

I have read and agree with the venue's withdrawal policy on behalf of myself and my co-authors.